# Radical Dehalogenation and Purine Nucleoside Phosphorylase *E. coli*: How Does an Admixture of 2′,3′-Anhydroinosine Hinder 2-fluoro-cordycepin Synthesis

**DOI:** 10.3390/biom11040539

**Published:** 2021-04-07

**Authors:** Alexey L. Kayushin, Julia A. Tokunova, Ilja V. Fateev, Alexandra O. Arnautova, Maria Ya. Berzina, Alexander S. Paramonov, Olga I. Lutonina, Elena V. Dorofeeva, Konstantin V. Antonov, Roman S. Esipov, Igor A. Mikhailopulo, Anatoly I. Miroshnikov, Irina D. Konstantinova

**Affiliations:** 1Shemyakin and Ovchinnikov Institute of Bioorganic Chemistry RAS, Miklukho-Maklaya 16/10, 117997 GSP, B-437 Moscow, Russia; kaushin@mail.ibch.ru (A.L.K.); julia.tok@mail.ru (J.A.T.); ifateev@gmail.com (I.V.F.); 8818818@mail.ru (A.O.A.); berzina_maria@mail.ru (M.Y.B.); a.s.paramonov@gmail.com (A.S.P.); olgusha2949@gmail.com (O.I.L.); iegol2013@gmail.com (E.V.D.); antonov.kant@yandex.ru (K.V.A.); esipov@ibch.ru (R.S.E.); aiv@mail.ibch.ru (A.I.M.); 2Institute of Bioorganic Chemistry, National Academy of Sciences, Acad. Kuprevicha 5/2, 220141 Minsk, Belarus; imikhailopulo@gmail.com

**Keywords:** purine nucleoside phosphorylase, biocatalyze, 3′-deoxyinosine, 2-fluorocordycepin, deuterium oxide

## Abstract

During the preparative synthesis of 2-fluorocordycepin from 2-fluoroadenosine and 3′-deoxyinosine catalyzed by *E. coli* purine nucleoside phosphorylase, a slowdown of the reaction and decrease of yield down to 5% were encountered. An unknown nucleoside was found in the reaction mixture and its structure was established. This nucleoside is formed from the admixture of 2′,3′-anhydroinosine, a byproduct in the preparation of 3-′deoxyinosine. Moreover, 2′,3′-anhydroinosine forms during radical dehalogenation of 9-(2′,5′-di-*O*-acetyl-3′-bromo- -3′-deoxyxylofuranosyl)hypoxanthine, a precursor of 3′-deoxyinosine in chemical synthesis. The products of 2′,3′-anhydroinosine hydrolysis inhibit the formation of 1-phospho-3-deoxyribose during the synthesis of 2-fluorocordycepin. The progress of 2′,3′-anhydroinosine hydrolysis was investigated. The reactions were performed in D_2_O instead of H_2_O; this allowed accumulating intermediate substances in sufficient quantities. Two intermediates were isolated and their structures were confirmed by mass and NMR spectroscopy. A mechanism of 2′,3′-anhydroinosine hydrolysis in D_2_O is fully determined for the first time.

## 1. Introduction

Among the modified nucleosides, 3′-deoxyribonucleosides remain very promising objects to study. The series of their research began with cordycepin—3′-deoxyadenosine (**1**)—the natural antimetabolite of adenosine (**2**) (Figure 1). Cordycepin was isolated from extracts of *Cordyceps millitaris* tissues in 1950 year [1,2]. This drug is an antioxidant and has a pronounced immunomodulatory effect. In the medicine of Southeast Asia (China, Japan, and Korea), Cordyceps extract has been used since ancient times as a remedy for tumors, viral and bacterial infections [3,4,5,6,7]. Currently, cordycepin is produced commercially in China using the *Chinensa militaris* [4,8].

Synthetic C2-substituted analogs of cordycepin (**3**, **4**) have a very interesting biological activity that may be applied in medicine. One of the best-known analogs of cordycepin is 2-fluorocordycepin (2-F-Cord, **3**) (Figure 1) [9,10]. This compound is currently being actively studied and tested as an antitumor and antiviral drug. The introduction of a chlorine or fluorine atom into a heterocycle of a nucleoside cardinally changes the biological properties of the molecule: pharmacokinetic properties, tissue distribution, metabolic pathway, and rate, as well as pharmacodynamics and cytotoxicity [11,12,13,14].

Recent studies have shown that 2-fluorocordycepin (**3**) is more efficient than cordycepin, in the treatment of human *African trypanosomiasis* [9,10]. Hence, creating effective ways to synthesize 2-fluorocordycepin (**3**) is a good direction of research.

It is known that 3′-deoxynucleosides can be synthesized using nucleoside phosphorylases (transglycosylation reaction). In our laboratory, an efficient method for obtaining of purine nucleoside phosphorylase was developed [15], and we used this enzyme in transglycosylation reactions.

However, the problem of a possible source of 3-deoxyribose arises [16,17]. Therefore, it is necessary to determine the optimal 3-deoxyribose donor among purine nucleosides. Recent industrial developments have made cordycepin available for use as a source of 3-deoxyribose. However, 3′-deoxyinosine (3′-dIno, **9**) is likely to perform better as a substrate in the synthesis of new nucleosides.

To test which compound could be a better donor of 3-deoxyribose, we recently implemented a modified chemical method for 3′-deoxyinosine synthesis from commercially available inosine (**5**) (Scheme 1) [18]. The details of this synthesis will be discussed later.

The first trial syntheses of 2-fluorocordycepin using purine nucleoside phosphorylase (PNP) (Scheme 2) yielded encouraging results: the conversion of 2-fluoroadenosine into the target product reached 67% after 7 days from the start of the reaction. In this synthesis, 2-fluoroadenosine (2-F-Ado) rather than 2-fluoroadenine (2-F-Ade) was used as the base donor. This choice was dictated by the extremely low solubility of 2-fluoroadenine in water (less than 0.5 mM at 50 °C) [18].

However, when we scaled the synthesis up to hundreds of milligrams of 2-fluorocordycepin, an unexpected slowdown of the reaction and a decrease in the conversion of 2-F-Ado into 2-fluorocordycepin down to 5% were encountered. Although the PNP worked normally, the mixture contained an unexpectedly small quantity of 3-deoxyribose phosphate (judging by the quantity of hypoxanthine, 5.45%), while 2-fluoroadenine (2-F-Ade) was the main product of the reaction (Appendix A). In addition, another nucleoside was found in the reaction mixture (Appendix A, RT = 4.802 min), [M + H]^+^ = 241.0923 (Appendix A).

Therefore, it was crucial to determine what disrupts the normal course of the transglycosylation reaction, what is the nature of the byproduct, and whether it is associated with the work of PNP.

## 2. Materials and Methods

### 2.1. General Procedures

Unless otherwise noted, the materials were obtained from commercial suppliers and used without any purification. D_2_O was obtained from Sigma-Aldrich. 2′,5′-Diacetyl-3′-deoxy-3′-xylobromoinosine **7** was synthesized as described in [18]. Recombinant *E.coli* PNP was obtained as described in [15].

Analytical HPLC was performed on the Waters system (Waters 1525, Waters 2489, Breeze 2), column Nova Pak C_18_, 4.6 × 150 mm, 5 µm, flow rate 0.5 mL/min. Method (I): gradient H_2_O → 25% MeOH/H_2_O, 20 min. Method (II): gradient H_2_O → 50% MeOH/H_2_O, 10 min.

NMR spectra were recorded on Bruker Avance II 700 spectrometers (Bruker BioSpin, Rheinstetten, Germany) in DMSO-d6 at 30 °C. Chemical shifts in ppm (δ) were measured relative to the residual solvent signals as internal standards (2.50). Coupling constants (*J*) were measured in Hz.

Liquid chromatography-mass spectrometry was performed on an Agilent 6210 TOF LC/MS system (Agilent Technologies, Santa Clara, CA, USA).

UV spectra were recorded on Hitachi U-2900 spectrophotometer (Tokyo, Japan).

### 2.2. Synthesis of Nucleosides

#### 2.2.1. Synthesis of 2′,5′-Di-O-acetyl-3′-deoxyinosine (**8**)

A flow of hydrogen was passed through a suspension of Pd/C (2.67 g, 10% wt % loading) in 11 mL of MeOH up to saturation of the catalyst with hydrogen. 5 g of CaCO_3_ were added; in 1 h, a solution of compound **7** (5 g, 12 mmol) in 20 mL of MeOH was added, and hydrogenation was continued. The reaction progress was monitored by TLC on silica gel (EtOAc–EtOH, 3:2). In 10 h, the reaction mixture was filtered off on Zeolite filter, the precipitate was washed by MeOH (50 mL), the filtrate was concentrated, and the target compound was isolated by chromatography on silica gel (column 2 × 17 cm, elution by a gradient of MeOH in CHCl_3_ (0% → 8%), 500 mL, flow rate 5 mL/min). Yield 2.80 g (8.3 mmol, 67%), white foam. All the MS and NMR data were identical to those described earlier [18].

#### 2.2.2. Synthesis of 3′-Deoxyinosine (**9**)

Aq. ammonia (25%; 56 mL) was added to a solution of diacetate **8** (2.8 g, 8.4 mmol) in MeOH (28 mL) and the reaction mixture was stirred at rt. After 3 h the deacetylation was complete [TLC, CHCl_3_–MeOH, 9:1 (vol)], the mixture was evaporated to dryness and the target product was isolated by a column chromatography on Separon (1.8 × 25 cm, gradient elution with MeOH in water, 0 → 30%, 2.0 L, flow rate 6 mL/min). Yield 1.62 g (6.45 mmol, 77%) as lyophilized powder; purity 99.12% (*R*_t_ = 6.947 min, method II). UV (H_2_O, pH 7.0) λ_max_, nm (ε, M^−1^cm^−1^): 248 (12,100); λ_min_, nm: 224 (4,500). All the MS and NMR data were identical to those described earlier [18].

#### 2.2.3. Synthesis of 2′,3′-Anhydroinosine (**11**)

Compound **7** (865 mg, 2.08 mmol) was dissolved in 16 mL of methanol, and 4 mL of 25% aqueous NH_4_OH was added to the solution. After 48 h incubation at room temperature, the solution was evaporated, and the residue was dried over P_2_O_5_. Yield 325 mg. The target compound was purified by recrystallization from water, yield 250 mg (1.00 mmol, 48%), RT 13.50 min, purity 96.5%. MS- and NMR-spectra data are provided in Appendix A.

HRMS (ESI^+^): (*m*/*z*): [M + H]^+^ calculated for C_10_H_11_N_4_O_4_ 251.0780, found 251.0765, [M + Na]^+^ calculated for C_10_H_10_N_4_NaO_4_ 273.0600, found 273.0581. Mp 230–232 °C. ^1^H NMR (700 MHz, DMSO-d6, 30 °C): *δ* = 12.39 (s, 1H, NH), 8.29 (s, 1H, H-8), 8.09 (s, 1H, H-2), 6.18 (d, *J*~1.0 Hz, 1H, H-1′), 5.03 (br. t, *J* = 4.85 Hz, 1H, OH-5′), 4.46 (d, *J* = 2.3 Hz, 1H, H-2′), 4.20 (d, *J*~2.0 Hz, 1H, H-3′), 4.19 (t, *J*~5.0 Hz,1H, H-4′), 3.53 ppm (m, 2H, CH_2_). ^13^C NMR (176 MHz, DMSO-d6, 30 °C): *δ* = 156.48 (C=O), 147.97 (C-4), 145.64 (C-2), 138.75 (C-8), 124.07 (C-5), 81.73 (C-1′), 80.94 (C-4′), 60.45 (C-5′), 58.16 (C-3′), 57.39 ppm (C-2′). ^15^N NMR (71 MHz, DMSO-d6, 30 °C): *δ* = 249.80 (N-7), 214.61 (N-3), 176.46 (N-9), 175.09 ppm (N-1).

#### 2.2.4. Compounds **10**, **12**, **14**

Compound **11** (58 mg, 0.23 mmol) and KH_2_PO_4_ (20 mg, 0.15 mmol) were dissolved in D_2_O (30 mL) (apparent pH 4.1) and incubated at 50 °C. In 96 h the solution was evaporated up to 3 mL, centrifuged, and the desired substances were isolated by HPLC using PerfectSil Target column (MZ-Analysentechnik, Germany), 20 × 250 mm, flow rate 4 mL/min, gradient elution with water-methanol system, 7–15%, 2 h. Yield: **10**—10 mg (0.04 mmol, 17%), **12**—5 mg (0.02 mmol, 9%), **14**—8 mg (0.03 mmol, 13%). The NMR spectrum data are provided in Appendix A.

### 2.3. Escherichia coli Purine Nucleoside Phosphorylase Inhibition Assay

Each reaction mixture (0.1 mL, 50 mM potassium phosphate, pH 7.0) contained 0.5 mM of the tested compound (**10**, **11**, **12**, or **14**), 0.5 mM nucleoside (inosine, 3′-deoxyinosine or 2-fluoroadenosine), and *Escherichia coli* purine nucleoside phosphorylase (0.0028 μg for inosine, 28 μg for 3′-deoxyinosine or 0.28 µg for 2-fluoroadenosine). Reaction mixtures were incubated at room temperature for 1 min. Substrate and product quantities were determined using HPLC (Waters 1525, column Nova-Pak C18, 4 μm, 3.9 × 150 mm, eluent 0.1% aqueous TFA, detection at 254 nm, Waters 2489).

### 2.4. Calculation of The Reaction Rate Constants and Solvent Kinetic Isotope Effects (KIE)

HPLC data of product **10** syntheses from epoxide **11** in H_2_O and D_2_O were used for the calculation of compound **10**, **11**, **12,** and **14** concentrations. Kinetic parameters were determined by nonlinear regression analysis using SciDAVis v1.D013 software. Three consecutive first- or pseudo- first-order reaction models were fitted:$$\begin{array}{c} A\overset{k_{1}}{\to }B\overset{k_{2}}{\to }C\overset{k_{3}}{\to }D\\ \left[A\right]=A_{0}\times e^{-k_{1}\times t}\\ \left[B\right]=A_{0}\times \frac{k_{1}}{k_{2}-k_{1}}\times \left(e^{-k_{1}\times t}-e^{-k_{2}\times t}\right)\\ \left[C\right]=A_{0}\times (\frac{k_{1}\times k_{2}}{\left(k_{2}-k_{1}\right)\times \left(k_{3}-k_{1}\right)}\times e^{-k_{1}\times t}+\frac{k_{1}\times k_{2}}{\left(k_{1}-k_{2}\right)\times \left(k_{3}-k_{2}\right)}\times e^{-k_{2}\times t}\\ +\frac{k_{1}\times k_{2}}{\left(k_{1}-k_{3}\right)\times \left(k_{2}-k_{3}\right)}\times e^{-k_{3}\times t})\\ \left[D\right]=A_{0}\times (\frac{k_{2}\times k_{3}}{\left(k_{2}-k_{1}\right)\times \left(k_{3}-k_{1}\right)}\times \left(1-e^{-k_{1}\times t}\right)+\frac{k_{1}\times k_{3}}{\left(k_{1}-k_{2}\right)\times \left(k_{3}-k_{2}\right)}\times \left(1-e^{-k_{2}\times t}\right)\\ +\frac{k_{1}\times k_{2}}{\left(k_{1}-k_{3}\right)\times \left(k_{2}-k_{3}\right)}\times \left(1-e^{-k_{3}\times t}\right)) \end{array}$$

## 3. Results

### 3.1. Deciphering the Mechanism of Degradation of Inosine Epoxide in D_2_O

As mentioned earlier, during the scaling up of 2-fluorocordycepin synthesis, we encountered an unexpected slowdown of the reaction and a decrease in the conversion of 2-F-Ado into 2-fluorocordycepin down to 5%. Although the PNP worked normally, the main product of the reaction was 2-fluoroadenine (2-F-Ade, RT 8.655 min) (Appendix A). In addition, another nucleoside was found in the reaction mixture (Appendix A, RT = 4.802 min), [M + H]^+^ = 241.0923 (Appendix A). Six days after the reaction started the mixture contained 30% of this unknown nucleoside. This percentage did not change during further incubation.

The nucleoside was isolated by the column reversed-phase chromatography, yield 10.6 mg (0.04 mmol, 17%). The structural formula of this nucleoside (compound **10**) was deduced from NMR data: ^1^H, ^1^H-^1^H-COSY, ^1^H-^13^C-HSQC, ^1^H-^13^C-HMBC, ^1^H-^15^N-HSQC, and ^1^H-^15^N-HMBC spectra (Figure 2).

We supposed that nucleoside **10** is formed from some impurity not detected in our preparation of 3′-deoxyinosine. More careful analysis using LC-MS helped us detect an admixture of 2′,3′-anhydroinosine **11** in the preparation. The data of chromate-mass spectrometry are shown in Appendix A. The structure of compound **11** was confirmed by NMR data (see Appendix A).

We assumed that epoxide **11** forms during dehalogenation of protected nucleoside **7** by tributyltin hydride (Scheme 1, Route A). To check this assumption, we replaced a tributyltin hydride treatment by dehalogenation using H_2_/Pd (Scheme 1, Route B). The resulted compounds **8** and **9** were identical to those described earlier [18]. No traces of epoxide **11** were found in nucleoside **9**. Compound **9** obtained with the help of catalytic dehalogenation was used in the synthesis of 2-fluorocordycepin.

According to HPLC data, the reaction mixture did not contain nucleoside **10** (Appendix A). The conversion of **9** into 2-fluorocordycepin **3** was 71% in 16 days.

It is the admixture of compound **11** in the first samples of 3′-dIno **9** that reduced the yield of 2-fluorocordecypin in the enzymatic reaction.

We performed a series of reactions of 2-fluorocordecypin synthesis using different samples of 3′-dIno **9**. The results are shown in Figure 3.

It can be seen that in cases **b** and **c,** the transglycosylation reaction and 2-fluorocordecypin synthesis proceed abnormally. At the beginning of the process (in the first two days), there was almost no difference between the three reactions, then the speed of 2-F-Cord **3** accumulation decreased, and conversion did not reach the optimum. Additional portions of the enzyme did not improve the situation—the synthesis of 2-F-Cord **3** would not resume.

Moreover, 2-fluoroadenine was formed in the mixture almost exclusively, with only traces of 3-deoxyribose-1-α-phosphate (determined by the amount of hypoxanthine in the mixture). We have never encountered such selective PNP behaviour in nucleoside synthesis.

On the other hand, if the slowdown of 2-F-Cord **3** synthesis had begun on the 3rd day, it could be assumed that not epoxide **11** itself, but the products of its degradation violate the normal course of the enzymatic reaction. Perhaps one of those products is a PNP inhibitor, which hinders the normal operation of the enzyme. Therefore, we decided to identify the mechanism of transformation of epoxide **11** into nucleoside **10**, isolate the intermediates, and determine their PNP inhibition constants.

The formation of nucleoside **10** from 2′,3′-anhydroinosine was described more than 40 years ago [19]. However, the transformation was performed under relatively harsh conditions (0.1 M NaOH, 80 °C, 60 min). In [20], **10** was synthesized from 2′,3′-anhydroadenosine, an intermediate product was isolated, and a mechanism of **10** formation was suggested. Nonetheless, we believed that more than one intermediate might exist, so we decided to investigate the reaction more closely.

First of all, we modified the synthesis of epoxide **11**. Numerous approaches to this synthesis have been proposed. Most commonly, a xylochloro- or xylobromo-derivative of a corresponding ribonucleoside acts as a precursor, and cyclization is performed using a strong base (for example, sodium methoxide [19,20,21]) followed by neutralization of the base or Dowex 1x2 (OH^−^) [22]. It turned out that the treatment of 2′,5′-diacetyl-3′-deoxy-3′-xylobromoinosine (it was synthesized earlier; see [18]) by the solution of aqueous NH_4_OH in methanol followed by simple evaporation and crystallization gave the pure desired product **11** with satisfactory yield.

Compound **11** was stable in pure water at 50 °C during 72 hours. It was stable under those conditions after the addition of LiCl and NaCl. In 5 mM potassium-phosphate buffer (pH 7.0) at 50 °C, the reaction proceeds. The HPLC profile of the reaction mixture is shown in Appendix A. We detected peaks that correspond to starting material **11** (RT 13.288 min.), nucleoside **10** (RT 7.038 min.), and several extra products (RT 4.380, 7.923, and 11.188). We supposed that those peaks (compounds **12**–**14**) are peaks of the sought intermediates. However, the quantity of those intermediates was too small for isolating them from the reaction mixture. Most likely, the lifetime of the intermediates is too short, and we had to find a way to slow down the reaction and to accumulate the substances in sufficient quantities.

We decided that the simplest way to do this is using D_2_O instead of H_2_O. HPLC profile of the reaction mixture (5 mM potassium-phosphate buffer, appeared pH 7.0, 50 °C, D_2_O) is shown in Appendix A.

The situation improved significantly. The relative quantities of intermediates increased, and we had a chance to isolate and investigate them. However, at pH 4.1 (appeared pH because the reaction was performed in D_2_O), the results were even better (Appendix A).

The dynamics of nucleosides **10** and **12** formation during hydrolysis of **11** at pH 7.0 and 4.1 in H_2_O and D_2_O are shown in Figure 4 and Figure 5.

It can be seen that the optimal conditions for obtaining of **12** are D_2_O at appeared pH 4.1. So, we performed hydrolysis of **11** in D_2_O at appeared pH 4.1 and isolated intermediates **12** and **14** (See Appendix A). Then our goal was to identify the chain of transformation—which of intermediates forms from **11** primarily and which one is an immediate precursor of **10**.

We put three reactions: hydrolysis of **11**, hydrolysis of **12**, and hydrolysis of **14,** and measured the UV spectra of the reaction mixtures overtime. Individual spectra of **10**, **11**, **12**, and **14** are shown in Figure 6.

The set of UV spectra for hydrolysis of **11** is shown in Figure 7. The picture is somewhat complicated due to the overlay of spectra of individual components. However, it can be seen that, in 48 h (magenta line), some quantity of **12** (shoulder at 270 nm) and **14** (λ_max_ 310 nm) is formed.

The set of spectra obtained during hydrolysis of **12** is much more useful (Figure 8).

In 24 h after reaction started, noticeable amounts of **14** (λ_max_ 310 nm) are formed (blue line). The absorption at 254 nm (λ_max_ of **12**) decreases, and an additional absorption maximum at 274 nm (λ_max_ of **10**) is appears. In 48 h (magenta line), the amounts of both **14** and **10** increase, and the amount of **12** is negligible. In 96 h (navy line), the amount of **14** decreases, while the amount of **10** continues increasing. In 144 h (violet line), the amount of **14** is minimal—almost all **14** is converted into **10**. These data allowed us to assume, that **11** is converted into **12**, then **12** is converted into **14**, and, finally, **14** is converted into **10**.

The last set of spectra was obtained during hydrolysis of **14** (Figure 9). The absorption at 310 nm is permanently decreased while absorption at 274 nm is increased. This confirms, that **14** is a direct precursor of **10,** and a chain of transformation is **11**–**12**–**14**–**10**.

The structure of compounds **12** and **14** was deduced from NMR data: ^1^H- and ^1^H-^1^H-COSY, ^1^H-^13^C-HSQC, ^1^H-^13^C-HMBC, ^1^H-^15^N-HSQC, and ^1^H-^15^N-HMBC. Assignments of main signals for **12** are shown in Figure 10.

Integrals of C2 and C8 protons (8.16 ppm and 7.97 ppm) in ^1^H-NMR spectra of **12** proved to be 0.66 and 0.17 correspondingly (integral 1.00 is assigned to anomer proton—6.08 ppm) (Appendix A). This can be explained by the H—D exchange in those positions. In ^2^H-NMR spectra of **12,** two peaks at 8.16 and 7.97 ppm with corresponding integrals are observed (Appendix A).

^13^C-NMR data also confirm the presence of deuterium in compound **12** (Appendix A).

Signals of C2 and C8 are shifted upfield (143.94 → 143.68 and 134.39 → 134.16 ppm) and are split into triplets. Signals in NMR spectra of nucleoside **14** were assigned in the same way (Figure 11).

Compound **13** (see Appendix A) isolated from the reaction mixtures was found to be a purinyl heterocycle (according to NMR data, see page S-32); λ_max_ 250 nm, λ_min_ 221 nm.

The H → D exchange at C8 is not related to hydrolysis of epoxide **11**. After incubation of inosine in D_2_O at 50 °C (pH 4.1) for 4 days, 81% of protons at C8 are changed for deuterium.

Thus, we have fully deciphered the structures of 2′,3′-anhydroinosine opening intermediates in a phosphate buffer solution. A full description of the NMR spectra of compounds **10**, **12**, and **14** is provided in the Appendix A.

### 3.2. Escherichia Coli Purine Nucleoside Phosphorylase Inhibition Assay

After obtaining compounds **10**, **12**, and **14**, a series of experiments were performed to study PNP inhibition. As we mentioned above, the presence of epoxide **11** in the reaction mixture inhibits the synthesis of 2-fluorocordycepin. So, epoxide **11** or one of the products formed during hydrolysis of **11** can be an inhibitor of PNP.

In this case, they should inhibit a phosphorolysis of inosine, 3′-deoxyinosine, or 2-fluoroadenosine. The results of phosphorolysis in the presence of compounds **10**, **11**, **12**, and **14** are shown in Figure 12. The reactions conditions are provided in Materials and Methods.

It can be seen that neither compound **11** nor compounds formed from **11** affect *E. coli* PNP activity.

### 3.3. Calculation of the Reaction Rate Constants and Solvent Kinetic Isotope Effects (KIE)

The rate constants and kinetic isotope effects for the synthesis stages of every isolated compound were determined (Table 1). Three consecutive first- or pseudo first-order reaction models were used.

## 4. Discussion

During a series of syntheses of 2-fluorocordycepin from 3′-deoxyinosine catalyzed by *E. coli* PNP (Scheme 2), we encountered the problem of a decrease in the yield of the target product. At the same time, *E. coli* PNP worked normally: 2-fluoroadenine was produced, while phosphorolysis of 3′-deoxyinosine did not occur (according to HPLC data, there was practically no hypoxanthine in the reaction mixture). As a result of a thorough study of 3′-deoxyinosine quality, an impurity of 2′,3′-anhydroinosine was detected. Acyclic nucleoside (**10)** was produced from this impurity during the enzymatic reaction carried out at pH 7.0 and 50 °C.

This brought up two questions: 1) where the impurity of 2′,3′-anhydroinosine appears in 3-′deoxyinosine, and 2) how this impurity prevents the phosphorolysis of 3-′deoxyinosine to 3-deoxyribose-1-α-phopsphate in the active site of *E. coli* PNP.

Acyclic nucleoside **10** was first discovered more than 40 years ago as an impurity in the synthesis of adenine nucleosides. The degradation of 2′,3′-anhydroadenosine in hot water was investigated in detail in [20]. An intermediate product (**10**) with opened pyrimidine ring was isolated. After adding NaOH to the reaction mixture, compound **10** was obtained. It was shown that 2′,3′-anhydroinosine (**11**) is converted into **10** during the heating of the solution of **11** in 0.1 M NaOH, yet the intermediate products were not isolated [19].

In our synthesis of 3′-deoxyinosine, different processes and conditions were used. Nevertheless, we assumed that at the stage of radical dehalogenation of 3′-deoxy-3′-xylobrom-2′,5′-diacetylinosine (**7**), in addition to the target 3′-deoxy-3′-xylobrom-2′,5′-diacetylinosine (**8**), an impurity of 5′-acetyl-2′,3′-anhydroinosine is formed. This impurity is difficult to separate by chromatography on silica gel. After removing acetyl groups by ammonolysis 3′-deoxyinosine, samples contained epoxide **11** (up to 17%), detected by high-resolution LC-MS.

Thus, we determined the compound that hinders the normal synthesis of 2-fluorocordycepine. We replaced the stage of radical dehalogenation with a catalytic one (Scheme 1, route B). Using the “new” 3′-deoxyinosine, we performed several enzymatic syntheses of 2-F-Cord in high yields (up to 72%).

However, it was difficult to identify the compound preventing the normal functioning of *E. coli* PNP. It was definitely not 2′,3′-anhydroinosine itself since enzymatic synthesis began to slow down only on the 3rd day from the beginning of the enzymatic reaction. At this time, the original 2′, 3′-anhydroinosine was already transformed into at least three nucleosides of a different structure (according to LC-MS data). How could these nucleosides be isolated from reaction mixtures, and how do they affect the functioning of PNP? It would be valuable to find inhibitors of *E. coli* PNP.

Obviously, it is necessary study the degradation mechanism of 2′,3′-anhydroinosine in a buffer solution at 50 °C thoroughly and attempt to isolate the degradation products quantitatively. However, in H_2_O, the degradation proceeded rather quickly, and the content of intermediates did not exceed 3–5%.

For this reason, we decided to replace H_2_O with D_2_O. This should slow down all chemical processes and allow a larger amount of intermediates to be accumulated in the solution. Our assumption was fully confirmed: the amount of intermediate nucleosides in the reaction mixture increased from 3–5% to 15%. However, at pH 7.0, the formation of the acyclic compound **10** proceeded rather quickly. We reduced the pH of the solution from 7.0 to 4.1; as a result, the rate of formation of **10** decreased by five times. In Appendix A, we provide the HPLC profile (Appendix A) of the inosine epoxide degradation reaction mixture at the optimum time (day 4) when the amount of intermediate degradation products is maximal. The conditions that allowed us to accumulate the degradation products in sufficient quantities were: **10**—10 mg (17%), **12**—5 mg (9%), **14**—8 mg (13%). The structure of these nucleosides was deduced from NMR spectra (see Appendix A).

It is clear that inosine epoxide ring opening in phosphate buffer is a multi-stage process. Our goal was to identify the chain of transformation - which one of the intermediates forms from **11** primarily and which one is an immediate precursor of **10**.

UV-spectroscopy helped us to understand this process. Each of the isolated nucleosides has own absorption maximum: nucleoside **10**—276, **11**—247, **12**—254, and **14**—310 nm. We investigated the change of UV spectra of reaction mixtures upon incubation of aqueous solutions for 0 to 312 h. Analysis of the dynamics of the spectra allowed us to determine the transformation chain of the inosine epoxide **11**: chain of transformation is **11**–**12**–**14**–**10**.

The reaction proceeds through the formation of intermediates **12** and **14**. A possible mechanism of compound 10 formation in D_2_O is shown in Scheme 3.

Performing the reaction in D_2_O allowed us to localize the proton positions and deuterium addition. This is useful for transient state and intramolecular rearrangements visualization.

The rate constants and kinetic isotope effects for the synthesis stages of each isolated compound were determined (Table 1).

It can be seen that the fastest stage is the synthesis of intermediate **14,** while the slowest one is the synthesis of cyclonucleoside **12**. At pH 4.1, the first stage is 1.4 times slower, and the next two stages are 3-4 times slower than at pH 7.0. The kinetic isotope effect of the solvent in the second stage is greater than in the first. In the synthesis of compound **10**, the reverse isotopic effect is observed.

Neither compound **11** nor compounds formed from **11** affect *E. coli* PNP activity (Figure 12). Therefore, the reason for the slowdown of transglycosylation reaction in the presence of **11** is still unclear. Probably some short-living intermediates interact with the active site, with their lifetime being too short for us to isolate them.

We believe that nucleosides with a structure similar to the compound shown in Scheme 3 could be effective inhibitors of *E. coli* PNP.

## 5. Conclusions

It is undesirable to use modified nucleosides synthesized with the help of radical dehalogenation (in particular, using Bu_3_SnH) as substrates of nucleoside phosphorylases. 2′,3′-Anhydronucleosides form during this reaction and the main product is difficult to separate from them. An admixture of 2′,3′-anhydroinosine was detected in some species of 3′-deoxyinosine synthesized using radical dehalogenation. It was found that derivatives of 2′,3′-anhydroinosine inhibit the forming of 1-α-phospho-3-deoxyribose during the synthesis of 2-fluorocordycepin analogs. A mechanism of hydrolysis of 2′,3′-anhydroinosine in D_2_O is fully determined for the first time. Two intermediates containing deuterium are isolated. The structure of those compounds was confirmed by UV-, mass-, and NMR-spectroscopy.

## Data Availability

The data presented in this study are available on request from the corresponding author.

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
