# Peer review of "Radical Dehalogenation and Purine Nucleoside Phosphorylase E. coli: How Does an Admixture of 2′,3′-Anhydroinosine Hinder 2-fluoro-cordycepin Synthesis"

_biomolecules, 2021, doi:10.3390/biom11040539_

Round 1

Reviewer 1 Report

In their manuscript Kayushin et al. present data on the enzymatic synthesis of the nucleoside analoga 2-fluoro-cordypin, the problems that occurred during the reaction as a consequence of impurities of the substrates. They investigated the nature of the contaminating substance, elucidated its degradation pathway and speculated that one of these intermediates might have PNP inhibitory activity. Moreover, experiments analyzing the cytotoxicity of 2-F-Cord have been conducted. However, the manuscript has several flaws.

General:

  • The manuscript appears to deal with two topics. On the one hand, the synthesis of 2-F-Cord and its putative use as an anti-cancer drug. On the other hand, the investigation of the nature of the contaminating substance, the intermediates that are produced during its degradation and the putative impact of the contaminating substance and/or its intermediates on the activity of the used E. coli PNP. I have the impression that putting both topics leads to a lack of focus. Therefore, I would suggest separating the manuscript into two more specific but clear papers dealing with each of the two subjects.

  • The English is terrible and needs excessive reworking, preferably by a native speaker.

  • Why are several sentences and single letters in the manuscript highlighted in yellow?

Major:

  • Compared to figure 6 figures 7, 8 and 9 have a lower quality. They lack sharpness. The quality of these figures should be improved. Moreover, in all four figures vertical lines indicating the absorption maxima for the single compounds should be included to facilitate understanding of the figure and to follow the argumentation of the authors in the text.

  • Line 357: I cannot follow this conclusion. On the one hand figures 12 and 13 unambiguously show that PNP activity is not affected by compounds 10, 11, 12 or 14. On the other hand, none of the experiments performed is suitable to study the interaction between 2-fluoroadenosine and the active site of the enzyme. The used experiments are only suitable to state weather the enzyme is active or not.” Even though the authors have reworded their statement it is still not correct. An activity assay is not suitable to study the interaction of a small molecule and the active site of a protein. A drop of enzymatic activity does not require binding to the active site. To verify an interaction between the compound and the active site other experiments are required.

  • Line 352ff: The figure legends for figures 12 and 13 have to be improved. Under which conditions was the activity assay conducted? How much of the putative inhibitor substances was added? How long was the assay conducted? Did you use different concentrations of the putative inhibitor?

  • Line 359: There is no subsection 3.1. included. So why using subsection 3.2 now? In general the results section should have more subsections dealing with a single aspect each.

  • Line 359ff: There is no experimental data and no figure for the cytotoxicity assay shown in the results section. A table with the values should be added and figure 14 definitely moved from the discussion to the results section.

  • Line 398: Really? Which one is it? What you found out is that there is an reduced activity when using a substrate prepared via radical dehalogenation as this is contaminated with compound 11 and that either this compound or whatever it is metabolized to might influence the activity of the E. coli PNP. However, it is still not clear which compound is responsible as seen in your figures 12 and 13.

  • Line 468: Why do you use however here? The data shown in fig. 14 did not suggest anti-tumour activity of 2-F-Cord (still about 80% cell viability at 50µM and about 75 % cell viability at 100 µM). Again a table showing cell viability values would be helpful.

  • Line 469: The statement of this sentence is not supported by an experiment or a reference. Either provide an experiment to support this statement or provide a suitable reference.

Minor:

Line 19: It is “ E. coli purine nucleoside phosphorylase”.

Line 60: its not “leukemia U937 and colon adenocarcinoma LS174T” but “leukemia U937 cells and colon adenocarcinoma LS174T cells”

Line 90/91: Why is “a small quantity of …” written in brackets

Line 170: “In parallel…” This is not a sentence.

Line 235: phosphate instead of phopspate

Line 328: The number of compound 12 should be written in bold.

Line 330: The number of compound 12 should be written in bold.

Line 447: The meaning of this sentence is not clear. Did compound 11 or the compounds formed from 11 affect the PNP activity or Did compound 11 or the compounds formed from 11 did not affect the PNP activity. Please clarify.

Line 453: There is a different font and/or font size used.

Line 471: “ADA” The abbreviation should be explained when appearing first

Line 474: Use µM instead of mkM.

Line 480: The graph in Fig. 14b says 20µM rather than 10µM.

Author Response

Review 1

General:

  • The manuscript appears to deal with two topics. On the one hand, the synthesis of 2-F-Cord and its putative use as an anti-cancer drug. On the other hand, the investigation of the nature of the contaminating substance, the intermediates that are produced during its degradation and the putative impact of the contaminating substance and/or its intermediates on the activity of the used E. coli PNP. I have the impression that putting both topics leads to a lack of focus. Therefore, I would suggest separating the manuscript into two more specific but clear papers dealing with each of the two subjects.

The experiments described in our article are devoted to solving synthetic problems of the biocatalytic method for producing 2-fluorocordycepin. The biological activity of this nucleoside is now being actively studied and not only antimicrobial, but also anticancer one. We present the first data on anticancer activity, which, in our opinion, may be very interesting and important not only for biochemists, but also for oncologists. Each reader will find interesting information for him in our article. Therefore, we ask you to publish our manuscript without splitting in two publications due to the relevance and importance of the information.

  • The English is terrible and needs excessive reworking, preferably by a native speaker.

Unfortunately, it is difficult to find in Moscow a native speaker, who is an expert in this field of science.

  • Why are several sentences and single letters in the manuscript highlighted in yellow?

The highlighting in yellow remained from revisions of the previous version of the article. They are removed now.

Major:

  • Compared to figure 6 figures 7, 8 and 9 have a lower quality. They lack sharpness. The quality of these figures should be improved. Moreover, in all four figures vertical lines indicating the absorption maxima for the single compounds should be included to facilitate understanding of the figure and to follow the argumentation of the authors in the text. Corrected (lines 335, 347, 354).
  • Line 357: I cannot follow this conclusion. On the one hand figures 12 and 13 unambiguously show that PNP activity is not affected by compounds 10, 11, 12 or 14. On the other hand, none of the experiments performed is suitable to study the interaction between 2-fluoroadenosine and the active site of the enzyme. The used experiments are only suitable to state weather the enzyme is active or not.” Even though the authors have reworded their statement it is still not correct. An activity assay is not suitable to study the interaction of a small molecule and the active site of a protein. A drop of enzymatic activity does not require binding to the active site. To verify an interaction between the compound and the active site other experiments are required.

The purpose of the experiments was to determine whether the putative inhibitors (comp. 10-14) affect the phosphorolysis of inosine, 3'-deoxyinosine, and 2-fluoroadenosine (we added Fig. 14, line 403).

In presence of the tested compounds, no slowdown of the reactions was found.If there were inhibitors among them, then the inhibition or dissociation constants would be determined for them.We believe that in this case, our experiments answer the question: do the degradation products of inosine epoxide isolated from the reaction mixture affect the synthesis of 2-fluorocordycepin.

  • Line 352ff: The figure legends for figures 12 and 13 have to be improved. Under which conditions was the activity assay conducted? How much of the putative inhibitor substances was added? How long was the assay conducted? Did you use different concentrations of the putative inhibitor?

The description of experiments is added to text (lines 170-171, 388-391).

Only one concentration of putative inhibitor was used (0.5 mM).

  • Line 359: There is no subsection 3.1. included. So why using subsection 3.2 now? In general the results section should have more subsections dealing with a single aspect each.

Corrected (lines 382, 409, 415)

  • Line 359ff: There is no experimental data and no figure for the cytotoxicity assay shown in the results section. A table with the values should be added and figure 14 definitely moved from the discussion to the results section.

 Experimental data, table, and figures 15a, 15b are added into Results section (lines 422-426).

  • Line 398: Really? Which one is it? What you found out is that there is an reduced activity when using a substrate prepared via radical dehalogenation as this is contaminated with compound 11 and that either this compound or whatever it is metabolized to might influence the activity of the E. coli PNP. However, it is still not clear which compound is responsible as seen in your figures 12 and 13.

Corrected (lines 509-515).

  •  Line 468: Why do you use however here? The data shown in fig. 14 did not suggest anti-tumour activity of 2-F-Cord (still about 80% cell viability at 50µM and about 75 % cell viability at 100 µM). Again a table showing cell viability values would be helpful.

A technical error was made when transferring information from Excel to OriginPro. We have corrected the data in Figure 15 (line 425). Experimental data on cell survival using different concentrations of 2-fluorocordycepin is added to the Results section (line 422).

  • Line 469: The statement of this sentence is not supported by an experiment or a reference. Either provide an experiment to support this statement or provide a suitable reference.

That just our assumption (line 536). We can delete it if you insist.

Minor:

Line 19: It is “ E. coli purine nucleoside phosphorylase” – Corrected (line 19).

Line 60: its not “leukemia U937 and colon adenocarcinoma LS174T” but “leukemia U937 cells and colon adenocarcinoma LS174T cells” – Corrected (lines 66-67).

Line 90/91: Why is “a small quantity of …” written in brackets

HPLC cannot register small quantities of 3-deoxyribose phosphate, but it can register small quantities of hypoxanthine which correlates with 3-deoxyribose phosphate quantities (line 97).

Line 170: “In parallel…” This is not a sentence. – Corrected (line 195).

Line 235: phosphate instead of phopspate – Corrected (line 261).

Line 328: The number of compound 12 should be written in bold. – Corrected.

Line 330: The number of compound 12 should be written in bold. – Corrected.

Line 447: The meaning of this sentence is not clear. Did compound 11 or the compounds formed from 11 affect the PNP activity or Did compound 11 or the compounds formed from 11 did not affect the PNP activity. Please clarify. – They do not affect. We added a Figure 14.

Line 453: There is a different font and/or font size used. – Corrected.

Line 471: “ADA” The abbreviation should be explained when appearing first. – Added to Abbreviations list (line 584).

 Line 474: Use µM instead of mkM. – Corrected everywhere.

Line 480: The graph in Fig. 14b says 20µM rather than 10µM. – This is an error on figure. Corrected (line 425).

Reviewer 2 Report

The submitted research work from Konstantinova group reports issues in enzymatic synthesis of 2-fluorocordycepin. This compound was previously synthesized by chemical and chemoenzymatic methods. Here authors used a published method for synthesis of 2-fluorocordycepin using purine nucleoside phosphorylase (PNP), 2-fluoroadenosine and 3’-deoxyinosine. They encountered issues because of the starting material of insufficient purity. The authors identified the impurity (2’,3’-anhydroinosine) and found out that this compound is hydrolyzed under the conditions of the enzymatic reaction. The authors identified and isolated products of its hydrolysis. However, none of the isolated compound had any inhibitory effect on purine nucleoside phosphorylase. Therefore, the mechanism of observed inhibition has not been elucidated. The authors added cytotoxicity data of 2-fluorocordycepin, 2-chlorocordycepin and cordycepin that are interesting by themselves but there is no link between these data and the rest of the manuscript.

The experiments are poorly described. The synthesis of 3’-deoxyinosine using tributyltin hydride was previously reported and the product was isolated as a pure compound. Here the synthesis was repeated, however no details are given (scale, yield). The authors suggest alternative synthesis of 3’-deoxyinosine, however they provide experimental data only to the first step (reductive debromination). Conditions for deprotection are not mentioned. There are no experimental details on PNP-catalyzed reaction of 2-fluoroadenosine with 3’-deoxyinosine contaminated with 2’,3’-anhydroinosine. Only a graph of 2-fluorocordycepin percentage in the reaction mixture (Figure 3) is provided with no additional information (scale, conditions, exact content of 2’,3’-anhydroinosine, additions of enzyme that are mentioned in the text). Later authors used convenient methods to identify products of 2’,3’-anhydroinosine hydrolysis and tested those derivatives for PNP inhibitory activity, however no inhibition was observed. Authors conclude that “derivatives of 2’,3’-anhydroinosine inhibit a forming of 1-α-phospho-3-deoxyribose during synthesis of 2-fluorocordycepins”, however they did not provide any direct evidence for this claim. I am afraid I cannot recommend this piece of work for publication in Biomolecules.

I have following questions:

What was the scale of 3’-deoxyinosine synthesis using tributyltin hydride, what was the purity of the final product and how was it determined?

What was the effect of compounds 10, 11, 12 and 14 on PNP activity for 2-fluoroadenosine phosphorolysis? Why no enzymatic syntheses of 2-fluorocordycepin in presence of compounds 10, 11, 12 and 14 were not performed?

How did authors prove that 3’-deoxyinosine is free of stannane contaminants that could also serve as inhibitors of the enzymatic reaction?

What is the structure of purinyl heterocycle 13? Its characterization data should be given. What is its effect on PNP activity?

The manuscript contains also several formal mistakes that complicate reading and understanding of the text:

Compounds are often mentioned in the text only as abbreviations (Ino, 2-F-Cord, 3’-dIno) without compound numbers. Compound number should be given in all cases the compound is mentioned.

“mkM” unit is used instead of µM.

There is a nitrogen missing in structure of compound 6 in Scheme 1.

In Scheme 1 conditions for hydrolysis of 11 are not given.

Line 131: Unit is missing in melting point data.

Line 170: 5-Fluorouracil is mistyped.

Line 170-171: Incomplete sentence “In parallel,…”

Figure 3, 4 and 14: Axes should be labeled: time (h), 2-F-Cord/nucleoside content (%)

A scheme of synthesis of epoxide 11 should be given.

Spectra of compound 10 should be omitted in figures 7-9.

Line 336: reference to Figures should be corrected.

Scheme 3: Mechanisms are usually drawn with curved arrows. The scheme should be redrawn accordingly. Protonation of epoxide 1 should contain curved arrow in opposite direction.

Table 2 and related text should be moved into results section.

Line 451: reference to Figure 3 should be corrected.

Reference formatting does not follow rules of the journal.

Author Response

Reviewer 2

The submitted research work from Konstantinova group reports issues in enzymatic synthesis of 2-fluorocordycepin. This compound was previously synthesized by chemical and chemoenzymatic methods. Here authors used a published method for synthesis of 2-fluorocordycepin using purine nucleoside phosphorylase (PNP), 2-fluoroadenosine and 3’-deoxyinosine. They encountered issues because of the starting material of insufficient purity. The authors identified the impurity (2’,3’-anhydroinosine) and found out that this compound is hydrolyzed under the conditions of the enzymatic reaction. The authors identified and isolated products of its hydrolysis. However, none of the isolated compound had any inhibitory effect on purine nucleoside phosphorylase. Therefore, the mechanism of observed inhibition has not been elucidated. The authors added cytotoxicity data of 2-fluorocordycepin, 2-chlorocordycepin and cordycepin that are interesting by themselves but there is no link between these data and the rest of the manuscript.

The experiments are poorly described. The synthesis of 3’-deoxyinosine using tributyltin hydride was previously reported and the product was isolated as a pure compound. Here the synthesis was repeated, however no details are given (scale, yield). The authors suggest alternative synthesis of 3’-deoxyinosine, however they provide experimental data only to the first step (reductive debromination). Conditions for deprotection are not mentioned. There are no experimental details on PNP-catalyzed reaction of 2-fluoroadenosine with 3’-deoxyinosine contaminated with 2’,3’-anhydroinosine. Only a graph of 2-fluorocordycepin percentage in the reaction mixture (Figure 3) is provided with no additional information (scale, conditions, exact content of 2’,3’-anhydroinosine, additions of enzyme that are mentioned in the text). Later authors used convenient methods to identify products of 2’,3’-anhydroinosine hydrolysis and tested those derivatives for PNP inhibitory activity, however no inhibition was observed. Authors conclude that “derivatives of 2’,3’-anhydroinosine inhibit a forming of 1-α-phospho-3-deoxyribose during synthesis of 2-fluorocordycepins”, however they did not provide any direct evidence for this claim. I am afraid I cannot recommend this piece of work for publication in Biomolecules.

I have following questions:

What was the scale of 3’-deoxyinosine synthesis using tributyltin hydride, what was the purity of the final product and how was it determined?

– The scale was nearly 300 mg. Purity (98.6%) was checked by HPLC.

What was the effect of compounds 10, 11, 12 and 14 on PNP activity for 2-fluoroadenosine phosphorolysis? Why no enzymatic syntheses of 2-fluorocordycepin in presence of compounds 10, 11, 12 and 14 were not performed? – Figure 14 with corresponding results is added (lone 403). The enzymatic synthesis of 2-fluorocordicepin in the presence of compounds 10, 11, 12, 14 took place, but with a very low yield. So far, we have not been able to isolate an effective synthesis inhibitor. We suggest that short-lived inosine epoxide metabolites may competitively inhibit PNP.

How did authors prove that 3’-deoxyinosine is free of stannane contaminants that could also serve as inhibitors of the enzymatic reaction? – We didn't checked if 3’-deoxyinosine is free of stannane contaminants. But after dehalogenation of compound 7 nucleoside 8 was isolated using chromatography on silicagel. Then ammonolysis was performed and the final product was isolated by reverse-phase chromatography. It is slightly possible that contaminants passed through all those stages. Moreover, PNP is active in synthesis of 2-F-Cord, as can be seen on Figure SI-1, but instead of target product a large quantity of 2-fluoroadenine (RT 8.65 min) is synthesized.

What is the structure of purinyl heterocycle 13? Its characterization data should be given. What is its effect on PNP activity? – Only 0.3 mg of this compound was isolated so we couldn't investigate its effect on PNP activity. NMR-spectra is added to SI (page S-32).

The manuscript contains also several formal mistakes that complicate reading and understanding of the text:

Compounds are often mentioned in the text only as abbreviations (Ino, 2-F-Cord, 3’-dIno) without compound numbers. Compound number should be given in all cases the compound is mentioned. – Corrected everywhere.

“mkM” unit is used instead of µM. – Corrected everywhere.

There is a nitrogen missing in structure of compound 6 in Scheme 1. – Corrected (line 84).

In Scheme 1 conditions for hydrolysis of 11 are not given. – Corrected (line 84).

Line 131: Unit is missing in melting point data. – Corrected (line 152).

Line 170: 5-Fluorouracil is mistyped. – Corrected (line 194).

Line 170-171: Incomplete sentence “In parallel,…” – Corrected (line 195).

Figure 3, 4 and 14: Axes should be labeled: time (h), 2-F-Cord/nucleoside content (%) – Corrected (lines 250, 302, 309, 425).

A scheme of synthesis of epoxide 11 should be given. – The scheme is added to SI (page S-3).

Spectra of compound 10 should be omitted in figures 7-9. – Compound 10 is a final product of all transformations. We believe that it should be shown on those diagrams.

Line 336: reference to Figures should be corrected. – Corrected (Line 374).

Scheme 3: Mechanisms are usually drawn with curved arrows. The scheme should be redrawn accordingly. Protonation of epoxide 1 should contain curved arrow in opposite direction. – Corrected (line 527).

Table 2 and related text should be moved into results section. – Corrected.

Line 451: reference to Figure 3 should be corrected. – Corrected (line 515).

Reference formatting does not follow rules of the journal. – Corrected.

Reviewer 3 Report

A comprehensive study regarding the formation of 2’,3’-anhydronucleosides is described based on a systematic study including experimental observations from a previous article disclosed by the same authors (reference 8), efforts to identify by-products, mechanistic studies, product isolation and full characterization, and finally the development of cytotoxicity studies.

Studying a purine nucleoside phosphorylase-catalyzed reaction, it seems that the product of some unexpected products is obtained, which is finally attributed due to the use of a synthetic step using Bu3SnH in the synthesis of the starting material. This finding turned on the possibility of forming a series of by-products. The identification of these new nucleosides suppose a nice achievement, deserving the article publication after the following modifications:

  • Line 116. Better use “of compound 7 (5 g, 12 mmol)” rather than “5 g (12 mmol) of compound 7
  • Line 120: “the gradient of MeOH in CHCl3” is not clear. Add initial and final percentage of each eluent solvent.
  • Line 139 (section 2.2.23): it is unclear for me the mass balance and yields, since only 23 mg and 39% yields is obtained, what about the missing material (other non identify products, starting material…). Please make a comment on this.
  • Line 195: indicate the retention time if 2-F-Ade: “(2-F-Ade, RT 7.038 min)” instead of “(2-F-Ade)”
  • Line 210: “We assumed that…” instead of “We decided that”
  • Line 210: “between the three reactions” instead of “between three reactions”
  • Line 274, 293 and others: “is shown in Figure x” instead of “is shown on Figure x”
  • Line 283: “…are those observed when the reaction was carried out in D2O at pH 4.1” instead of “…are performing of reaction in D2O at appeared pH 4.1”
  • Line 295: “say in 48 h”? Revise this part of the sentence.
  • Line 390: for clarity, move reference 10 to the end of the sentence
  • Reference section: use abbreviation for journal names in reference 1b, 2d, 2e, 9, 15b and 15c
  • Reference section: Delete issue numbers in references 14b, 15b, 15c and 15d

Overall, the article is well executed and might be of interest for a broad audience in Biomolecules.

Author Response

Reviewer 3

A comprehensive study regarding the formation of 2’,3’-anhydronucleosides is described based on a systematic study including experimental observations from a previous article disclosed by the same authors (reference 8), efforts to identify by-products, mechanistic studies, product isolation and full characterization, and finally the development of cytotoxicity studies.

Studying a purine nucleoside phosphorylase-catalyzed reaction, it seems that the product of some unexpected products is obtained, which is finally attributed due to the use of a synthetic step using Bu3SnH in the synthesis of the starting material. This finding turned on the possibility of forming a series of by-products. The identification of these new nucleosides suppose a nice achievement, deserving the article publication after the following modifications:

  • Line 116. Better use “of compound 7 (5 g, 12 mmol)” rather than “5 g (12 mmol) of compound 7” – Corrected (lines 134, 142).
  • Line 120: “the gradient of MeOH in CHCl3” is not clear. Add initial and final percentage of each eluent solvent. – Corrected (line 139).
  • Line 139 (section 2.2.23): it is unclear for me the mass balance and yields, since only 23 mg and 39% yields is obtained, what about the missing material (other non identify products, starting material…). Please make a comment on this.

– Please take a look at Figure SI-7. You can see that a lot of starting material (epoxide 11) is left in the reaction mixture. If we incubate the reaction for a longer time, the content of intermediates decreases. That's why we stopped the reaction in 4 days and that's why the total yield is low.

  • Line 195: indicate the retention time if 2-F-Ade: “(2-F-Ade, RT 7.038 min)” instead of “(2-F-Ade)” – Corrected (line 233)
  • Line 210: “We assumed that…” instead of “We decided that” – Corrected (line 250).
  • Line 210: “between the three reactions” instead of “between three reactions” – Corrected (line 273)
  • Line 274, 293 and others: “is shown in Figure x” instead of “is shown on Figure x” – Corrected everywhere.
  • Line 283: “…are those observed when the reaction was carried out in D2O at pH 4.1” instead of “…are performing of reaction in D2O at appeared pH 4.1” – Partially corrected. We kept "appeared pH". The reaction was performed in D2O, so the term "pH" cannot be used as is.
  • Line 295: “say in 48 h”? Revise this part of the sentence. – Corrected.
  • Line 390: for clarity, move reference 10 to the end of the sentence – Corrected.
  • Reference section: use abbreviation for journal names in reference 1b, 2d, 2e, 9, 15b and 15c – Corrected everywhere.
  • Reference section: Delete issue numbers in references 14b, 15b, 15c and 15d – Corrected.

Round 2

Reviewer 1 Report

After revision, the quality of the manuscript has been significantly improved. The majority of my comments were taken into account and most of my questions were answered. Especially the quality of the presented figures increased. Even if I still think that it would be better to split the manuscript into two more specific ones, I can respect and accept the authors' wish not to do so. However, I have still a few comments that might help to improve the manuscripts.

General:

The lack of native speakers in Moscow is not an excuse for not being able to revise English. In the Email age it is not necessary that the person who does the revision is located in Moscow. Even MDPI offers an English editing service for authors. If a suitable native speaker is not available, such a service should be used before the manuscript can be published.

Major:

  • The comment (“Line 357… [now line 431]) has still not been answered satisfactorily. I suspect that the authors do not understand what I mean.
    1. The sentence contains a double negative.While figures 12-14 clearly show that none of the compounds (10-14) significantly affect the activity of the enzyme, the sentence expresses by the double negative "...neither compound...should not affect..." that all compounds affect the activity, which is obviously in contrast to the figures.
    2. The authors claim that there is an “…interaction with the active site…” but this is not supported by the experiment. The experiment support whether or not there is enzymatic activity. But the experiment did not support whether or not there is an interaction of the compound with the enzyme.

This has to be corrected and clarified.

  • I requested to improve the figure legends of Fig. 12 and 13. It is nice that the information was added in the Materials and Methods section, however, the figure legend has still not improved. Either state that the experiment was conducted as described in the Materials and Methods section or mention the conditions in the Fig. legend and not results section plain text.

  • The subsections in the results section need to be redesigned. The first subsection (3.1.) is not "Escherichia coli purine nucleoside phosphorylase inhibition assay" but whatever is described directly after the section "3. results".

Minor:

  • The implementation of the vertical lines in Fig. 7-9 has improved understandability of the spectra significatntly. Nevertheless, I would suggest that the same vertical lines are included in Fig. 6 and that these lines are labelled in all figures (6-9) with the corresponding wavelength rather that labelling the peaks as in Fig. 6.

Author Response

Reviewer 1, Round 2.

After revision, the quality of the manuscript has been significantly improved. The majority of my comments were taken into account and most of my questions were answered. Especially the quality of the presented figures increased. Even if I still think that it would be better to split the manuscript into two more specific ones, I can respect and accept the authors' wish not to do so. However, I have still a few comments that might help to improve the manuscripts.

General:

The lack of native speakers in Moscow is not an excuse for not being able to revise English. In the Email age it is not necessary that the person who does the revision is located in Moscow. Even MDPI offers an English editing service for authors. If a suitable native speaker is not available, such a service should be used before the manuscript can be published.

English is corrected. 

Major:

  • The comment (“Line 357… [now line 431]) has still not been answered satisfactorily. I suspect that the authors do not understand what I mean.
    1. The sentence contains a double negative.While figures 12-14 clearly show that none of the compounds (10-14) significantly affect the activity of the enzyme, the sentence expresses by the double negative "...neither compound...should not affect..." that all compounds affect the activity, which is obviously in contrast to the figures.
    2. The authors claim that there is an “…interaction with the active site…” but this is not supported by the experiment. The experiment support whether or not there is enzymatic activity. But the experiment did not support whether or not there is an interaction of the compound with the enzyme.

Now it looks as follows: "It can be seen that neither compound 11 nor compounds formed from 11 affect E. coli PNP activity."  (Lines 381, 481).

  • I requested to improve the figure legends of Fig. 12 and 13. It is nice that the information was added in the Materials and Methods section, however, the figure legend has still not improved. Either state that the experiment was conducted as described in the Materials and Methods section or mention the conditions in the Fig. legend and not results section plain text.

The experiment conditions are removed and reference to Materials and Methods is added. (Line 368).

  • The subsections in the results section need to be redesigned. The first subsection (3.1.) is not "Escherichia coli purine nucleoside phosphorylase inhibition assay" but whatever is described directly after the section "3. results".

The subsection 3.1 added, other sections are renumerated.

Minor:

  • The implementation of the vertical lines in Fig. 7-9 has improved understandability of the spectra significatntly. Nevertheless, I would suggest that the same vertical lines are included in Fig. 6 and that these lines are labelled in all figures (6-9) with the corresponding wavelength rather that labelling the peaks as in Fig. 6.

Vertical lines are included in Fig 6; all lines are labelled.

Reviewer 2 Report

The updated manuscript has been improved in terms of formal requirements but the scientific part still requires major changes. Here are the most important issues:

The authors responded to my question regarding the purity of 3´-deoxyinosine saying that the product was over 98% pure. I am not convinced about that as Figure SI-3 shows 16 % of compound 11 and also some inosine.

The authors mentioned in their response that the enzymatic synthesis of 2-fluorocordicepin in the presence of compounds 10, 11, 12, 14 had took place, but with a very low yield. This should be mentioned in the article, corresponding graphs showing yields should be added.

Figure 3 has not been changed according to my suggestions (scale, conditions, exact content of 2’,3’-anhydroinosine, additions of enzyme that are mentioned in the text).

The vague conclusion of the article remained unchanged.

For these reasons, I cannot recommend this article for publication in Biomolecules.

Author Response

Reviewer 2, Round 2

The updated manuscript has been improved in terms of formal requirements but the scientific part still requires major changes. Here are the most important issues:

The authors responded to my question regarding the purity of 3´-deoxyinosine saying that the product was over 98% pure. I am not convinced about that as Figure SI-3 shows 16 % of compound 11 and also some inosine.

We said that the scale was nearly 300 mg. Purity (98.6%) was checked by HPLC.

However, the synthesis of 2-fluoroadenosine with this sample of 3'-deoxyinosine proceeded with low conversion of 2-fluoroadenosine to 2-fluorocordycepin. We checked quality of the starting nucleosides thoroughly: changed the HPLC system and the HPLC column. We detected an admixture epoxide in 3'-deoxyinosine sample (16% according to LC-MS). The data is shown in Figure SI-3.

The authors mentioned in their response that the enzymatic synthesis of 2-fluorocordicepin in the presence of compounds 10, 11, 12, 14 had took place, but with a very low yield. This should be mentioned in the article, corresponding graphs showing yields should be added.

The slowdown of 2-fluorocordycepin synthesis in presence of epoxide 11 is shown in Figure 3 (line b). The synthesis was carried out for more than 400 hours. In 72 hours all the compounds, which are converting into nucleoside 10, were present in the reaction mixture. We did not investigate the effect of each of them on synthesis of 2-fluorocordycepin separately. Moreover, this would be not a very correct experiment: the synthesis of 2-fluorocordycepin takes a long time while the degradation of the inosine epoxide proceeds relatively fast.

Figure 3 has not been changed according to my suggestions (scale, conditions, exact content of 2’,3’-anhydroinosine, additions of enzyme that are mentioned in the text).

Reaction conditions and epoxide 11 concentrations are added to legend (Line 242).

The vague conclusion of the article remained unchanged.

For these reasons, I cannot recommend this article for publication in Biomolecules.